# Role of Mitochondria in Interplay between NGF/TRKA, miR-145 and Possible Therapeutic Strategies for Epithelial Ovarian Cancer

**DOI:** 10.3390/life12010008

**Published:** 2021-12-21

**Authors:** Daniela B. Vera, Allison N. Fredes, Maritza P. Garrido, Carmen Romero

**Affiliations:** 1Laboratory of Endocrinology and Reproductive Biology, Clinical Hospital University of Chile, Santiago 8380456, Chile; dverap@hcuch.cl (D.B.V.); allison.fredes@ug.uchile.cl (A.N.F.); 2Obstetrics and Gynecology Departament, Faculty of Medicine, University of Chile, Santiago 8380453, Chile

**Keywords:** mitochondria, oxidative phosphorylation (OXPHOS), neurotrophins (NTs), nerve growth factor (NGF), tropomyosin receptor kinase A (TRKA), epithelial ovarian cancer (EOC), miRs, miR-145, chemoresistance, antitumoral complementary therapies

## Abstract

Ovarian cancer is the most lethal gynecological neoplasm, and epithelial ovarian cancer (EOC) accounts for 90% of ovarian malignancies. The 5-year survival is less than 45%, and, unlike other types of cancer, the proportion of women who die from this disease has not improved in recent decades. Nerve growth factor (NGF) and tropomyosin kinase A (TRKA), its high-affinity receptor, play a crucial role in pathogenesis through cell proliferation, angiogenesis, invasion, and migration. NGF/TRKA increase their expression during the progression of EOC by upregulation of oncogenic proteins as vascular endothelial growth factor (VEGF) and c-Myc. Otherwise, the expression of most oncoproteins is regulated by microRNAs (miRs). Our laboratory group reported that the tumoral effect of NGF/TRKA depends on the regulation of miR-145 levels in EOC. Currently, mitochondria have been proposed as new therapeutic targets to activate the apoptotic pathway in the cancer cell. The mitochondria are involved in a myriad of functions as energy production, redox control, homeostasis of Ca^+2^, and cell death. We demonstrated that NGF stimulation produces an augment in the Bcl-2/BAX ratio, which supports the anti-apoptotic effects of NGF in EOC cells. The review aimed to discuss the role of mitochondria in the interplay between NGF/TRKA and miR-145 and possible therapeutic strategies that may decrease mortality due to EOC.

## 1. Introduction

Ovarian cancer (OC) is one of the deadliest gynecological tumors worldwide, the seventh-most common cancer in women, and the eighth-most common cause of cancer death [1,2]. OC has been diagnosed in advanced stages due to a lack of specific symptoms and biomarkers that make it difficult for the early detection of the disease [3]. Five-year relative survival in patients with OC is below 45%, and the proportion of women who die from this disease has not improved substantially over time, unlike other common cancer types [4]. Standard treatment consists of cytoreductive surgery and, after, chemotherapy based on platinum and taxol compounds. Punctually, chemotherapy considers cycles of intravenous carboplatin plus paclitaxel [5,6]. In general, patients who receive therapy in early stages of OC tolerate the treatment well and go into remission, but cancer relapse is frequent [4,5]. Consequently, there is a need for new therapeutic approaches to prevent recurrence and avoid death by OC.

Among the different histological types of OC, the most common is epithelial ovarian cancer (EOC), representing almost 90% of ovarian malignancies [1,7]. EOC comprises several subtypes, such as serous, endometrioid, mucinous, clear cell, and mixed. High-grade serous ovarian cancer (HGSOC) is the most frequent histological subtype [7,8]. At the same time, cytoreductive surgery and chemotherapy are first-line therapies [8]. Eventually, the patients experience chemoresistance, which represents a problem and decreases the chances of eradicating cancer cells [5,9]. Unfortunately, current therapies are not as effective as expected because 70–80% of patients with OC suffer a relapse within the first 2 years [10]. Hence, it is essential to understand the physiopathology of this disease to develop new therapeutic strategies or enhance the existing ones. 

Cancer cells amass metabolic alterations that allow them to access unusual nutrient sources to sustain accelerated proliferation. More recent studies revealed that mitochondria play a crucial role in carcinogenesis by modulating cell proliferation and resistance to apoptosis in cancer cells [10,11]. Evidence showed that mitochondrial activity still participates in tumor energy production [12,13]. Hence, cancer cells adapt their metabolism to acquire energy from the nutrient-poor environment to survive and proliferate. Accordingly, these adaptations in the tumor cells favor them to survive and proliferate, a phenomenon known as metabolic reprogramming [11,12,13] and considered one of the hallmarks in cancer cells [14].

## 2. Metabolic Changes in Epithelial Ovarian Cancer Cells

Previous research has identified the mitochondria as a possible target that allows them to attack the cancer cell and activate apoptosis. The mitochondria are the metabolic center of the cell, generate ATP through oxidative phosphorylation (OXPHOS), the cycle of Krebs, β-oxidation of fatty acids, and participate in control redox by the production of reactive oxygen species (ROS), buffering of calcium, and death cell [13,15,16]. As an endosymbiotic organelle, the structure of mitochondria shows a double membrane separated by an intramembrane space [15,17]. The outer membrane surface is enriched in proteins for the import of nuclear-encoded proteins. The inner membrane has a larger surface than the outer membrane and contains many finger-like projections protruding into the matrix called cristae [18]. The matrix contains mitochondrial DNA (mtDNA), a genome comprising 37 genes that encode 13 polypeptides, 2 ribosomal RNA (rRNA), and 22 transference RNA (tRNA) [15]. Therefore, the alteration of structures or functions of mitochondria contributes to several diseases, including neurodegenerative, metabolic diseases, and cancer [14,19,20].

Interestingly, mitochondria participate in homeostatic processes and cell death, acting as a critical component of apoptosis and autophagy [21,22,23]. There are two apoptotic signaling pathways: the extrinsic pathway and the intrinsic or mitochondrial pathway of apoptosis [10]. The outer membrane permeabilization is driven by proapoptotic effector members of the B cell lymphoma 2 (Bcl-2) family of proteins, as BAX and BAK, that activate apoptotic signaling pathways and the release of cytochrome C (Cyt C), which triggers apoptosis in cancer cells [21,24].

The cancer cells exhibit metabolic changes compared with non-tumoral cells [14]. Tumor cells grow in harsh environments, with low availability of nutrients and oxygen, known as the tumoral microenvironment (TME) [25]. Cancer cells display increased glucose uptake, which generates lactate, a phenomenon known as the Warburg effect or aerobic glycolysis [12,26]. Otto Warburg described in 1942 that, in cancer cells, the main metabolic pathway used was glycolysis, independent of oxygen availability [26]. The Warburg effect triggers mechanisms that regulate the expression of transcriptional regulators such as Hypoxia-inducible factor-1 alpha (HIF-1α), tumor suppressor protein p53, or oncogene c-Myc [27]. Additionally, cancer cells overexpress glucose transporters (GLUT) and there is an increased uptake of glucose that is catabolized by the tumor under aerobic conditions [14]. For instance, ovarian tumors express high levels of GLUT1 that allow them to maintain an increased proliferative and survival potential [28,29,30]. 

Although the Warburg effect is described as one of the main metabolic pathways, it has been determined that different cancers use other pathways such as OXPHOS [31,32,33,34]. Studies have revealed apparent metabolic differences in cancer cell lines and different stages of OC in biopsies of patients (Table 1). Clinical studies in tissues of patients with OC showed higher glycolytic activity than in non-cancer ovarian tissues. Remarkably, the authors reported the increase in proteins involved in the final step of glycolysis, such as Pyruvate kinase M2 (PKM2) [35], various glycolytic genes that, through the expression of key glycolytic enzymes, show the Warburg effect, such as Hexokinase 1 (HK1) and Hexokinase 2 (HK2), among others, which are shown in Table 1 [36,37].

Additionally, there is an increase in glycolysis in advanced stages of EOC, as stage III or IV, compared to early stages, an increase that was also observed in serous versus non-serous carcinoma tissues [37,38]. In tumors representing HGSOC, some ovarian cancer cells were chemosensitive and others were chemoresistant to platinum therapy. The results with glycolysis inhibitors decreased the proliferation of cell lines, which shows the importance of the glycolytic pathway in the progression of OC and in the development of new therapies that aim to inhibit this metabolic pathway [38]. However, the evidence collected indicates that OXPHOS would have an essential role in EOC [39]. Tissue analysis of EOC shows an increase in Peroxisome Proliferator-Activated receptor-gamma coactivator-1 alpha (PGC-1α), a coactivator that stimulates mitochondrial biogenesis and regulates the synthesis of proteins involved in OXPHOS [40,41,42]. The above mentioned is consistent with other studies, where an increase in OXPHOS was observed in EOC compared to non-tumor tissue [43].

The implications of the increase in OXPHOS have been studied in patient samples and different OC cell lines. Thus, it was determined in HGSOC cancer cells that low levels of Tumor necrosis factor receptor-associated protein 1 (TRAP1) were associated with cell lines with predominantly oxidative metabolism, which promotes the progression and survival of ovarian cancer cells resistant to platinum therapy. Additionally, lower levels of TRAP1 were related with higher severity and lower progression-free survival in samples from patients with OC [44]. However, another report has established metabolic heterogeneity even in HGSOC, studied in tissue samples and ovarian cancer cell lines, where subgroups of metabolic profiles characterized high OXPHOS and low OXPHOS. In this way, cell lines with high OXPHOS displayed a high expression of electron transport chain (ETC) proteins, mitochondrial content, levels of ATP and ROS, more significant oxidative stress, oxygen consumption rate, and dysregulation in iron homeostasis, characteristics that would be favoring a better prognosis and sensitivity to therapy [45].

These antecedents show the importance of the compression of metabolic heterogeneity and, in particular, OXPHOS for developing new therapies that target the glycolytic mechanism and the mitochondrial metabolism, which plays a crucial role in various pathways and mechanisms related to tumorigenesis in OC.

**Table 1 life-12-00008-t001:** Metabolic changes in different models of epithelial ovarian cancer.

Pathways	Models of Study in EOC	Description of Evidence	References
Glycolysis	Tissue	Increase in Pyruvate kinase M2 (PKM2), inducible Nitric Oxide Synthase (iNOS), and glycolytic genes (i.e., SLC2A1, SLC2A4, HK1, HK2, PFKFB3, PDK3, and LDHA). Glycolytic metabolism was observed to increase in EOC cells in comparison with non-cancer cells.	[36]
	Tissue	The glycolytic enzyme HK2 is higher in EOC tissues than in normal ovarian tissues, in advanced stages, and serous carcinomas than in non-serous carcinomas.	[37]
	Tissue and cell lines	The expression of glycolytic proteins is higher in HGSOC and advanced stages of OC (III / IV) than in the early stages. Glycolysis inhibitors decrease the proliferation of HGSOC cell lines sensitive and resistant to therapy.	[38]
OXPHOS	Tissue	Increase in mtDNA, numbers of mitochondria, and levels of proteins associated with OXPHOS (i.e., PGC-1α).	[41]
	Tissue	Increased OXPHOS is relative to non-cancer ovarian tissue.	[43]
	Tissue and cell lines	Increased levels of TRAP1 were associated with higher OXPHOS, stage, resistance to therapy, and lower survival.	[44]
	Tissue, cell lines, Patient-derived xenografts (PDX)	HGSOC high in OXPHOS displays a better prognosis and sensitivity to chemotherapy.	[45]

Mitochondria are steadily undergoing fusion and fission to share organelle contents, allowing for mitochondrial networking, enabling quality control by regulating mitophagy, autophagy, and apoptosis [46]. Fusion is regulated by the proteins Mitofusin 1 (MFN1) and Optic atrophy 1 (OPA1), while the proteins Fission 1 (FIS1) and Dynamin-related protein 1 (DRP1) regulate fission [47] (Figure 1). 

Formerly, Grieco and coworkers reported changes in mitochondrial morphology with increasing malignity, namely, from network mitochondria to a single, enlarged mitochondrion in mouse ovarian surface epithelial (MOSE) cells [48]. The authors argue that these mitochondria changes may help adapt to hypoxia, and alterations accompany the modifications in the mitochondrial ultrastructure, mitochondrial membrane potential, and the adjustment of ROS levels [48]. Currently, mitochondria are proposed as one of the therapeutic targets to avoid chemoresistance [11,49]. Heretofore, Drp1 was examined as a prognostic biomarker for EOC. The authors examined the prognostic impact of Drp1 and its phosphorylated forms in EOC [50]. High expression levels of phosphorylated Drp1 were associated with significant malignancy and patients non-responsive to adjuvant chemotherapy [50].

Recently, the role of cancer stem cells (CSCs) in TME and their implications in chemoresistance in OC have been discussed [51]. CSCs are decidedly plastic to the TME; hence, these cells could be more resistant to chemotherapy [52]. Additionally, a few isolated CSCs would be involved in tumorigenesis and would be resistant to chemotherapy and radiation due to their dormant state and a high expression of drug efflux pumps. Consistently, CSCs can adapt to TME and survive due to their DNA repair mechanisms and their capacity to evade the immune system of oncologic patients [51,52].

## 3. Role of Nerve Growth Factor and microRNAs in EOC 

### 3.1. Role of Nerve Growth Factor in Epithelial Ovarian Cancer

In the ovary, neurotrophins (NTs) play an essential role in ovarian performance [53]. NTs belong to a family of growth factors that promote neuronal survival and differentiation and display crucial functions in non-neuronal cells, such as the ovary. They are involved in folliculogenesis, steroidogenesis, and angiogenesis [54,55,56,57,58]. In EOC biopsies, Nerve Growth Factor (NGF) and its high-affinity receptor tropomyosin kinase A (TRKA) increase during EOC progression [59]. The active form of the receptor (p-TRKA) showed the most significant increase in EOC biopsies, which suggests that p-TRKA could be a potential marker of malignancy of EOC [59]. In vitro and ex vivo experiments have shown that NGF/TRKA play a key role in EOC pathogenesis, promoting essential processes such as cell proliferation, invasion, migration, and angiogenesis by increasing several oncogenic proteins, such as vascular endothelial growth factor (VEGF) [58,59,60,61,62]. In this context, NGF acts as a direct and indirect angiogenic factor, promoting an increase of VEGF expression in EOC cells [59,60] and stimulating endothelial cells directly to enhance tumoral angiogenesis [63]. 

Angiogenesis involves releasing mitogenic growth factors to the endothelium, a process depending on a pro- and antiangiogenic balance [27,64]. Increased angiogenesis occurs in premalignant events that may lead to cancer [62]. Our previous studies using microarrays’ analysis found that EOC explants stimulated with NGF overexpress most genes related to cellular proliferation [65]. Additionally, global gene expression profiles of serous EOC human samples were obtained with DNA oligonucleotides microarray, resulting in a list of 22 genes related to various features of the NGF/VEGF signaling pathway. Therefore, functional enrichment analysis of upregulated genes identified predominant GO terms: apoptosis, NGF receptor signaling pathway, transcription factor activity, and steroid binding [65]. 

In turn, specific genes were evaluated in the NGF signaling involved in cellular proliferation, such as TRKA, PI3K, AKT2, MAPK, and FOXL-2. Additionally, aiming to assess the pro- or anti-apoptotic effect of NGF, the proteins Bcl-2 and BAX were determined. Both proteins regulate apoptosis downstream of the master regulator p53 [38]. Bcl-2 prevents apoptosis upon several stimuli, while BAX forms a heterodimer with Bcl-2, thus exerting a proapoptotic effect [66]. EOC cells are intrinsically resistant to cell death, and NGF stimulation produces an increase in the Bcl-2/BAX ratio, which supports the anti-apoptotic effects of NGF in EOC cells [65].

Similar to NGF, some researchers have described the pro-tumoral role of other NTs, such as brain derivate neurotrophic factor (BDNF) and its high-affinity receptor TRKB [67]. BDNF from follicular fluid stimulates the TRKB receptor of fallopian tube epithelium cells, promoting their survival, migration, and attachment, which are critical stages in the EOC tumorigenesis [68]. Direct stimulation with BDNF enhances cell proliferation, migration, invasion, and angiogenesis potential [69]. Besides, BDNF/TRKB could directly induce endothelial cell migration [69], indicating that NTs are a critical component in EOC angiogenesis. These studies show that NTs contribute to the tumoral progression of EOC, acting as autocrine growth factors and angiogenic factors.

On the other hand, NTs could contribute to other critical processes for cancer adaptation, such as increasing chemoresistance in EOC cells. NTs-mediated chemoresistance is favored by activation of key signaling pathways, such as the Wnt/β-catenin pathway [70]. This signaling pathway promotes the increase of epithelial–mesenchymal transition (EMT) proteins and favors EOC cells’ more migratory and chemoresistant phenotype [50]. In addition, NTs and their receptors could be involved in the modulation of efflux transporters in EOC cells. The tyrosine kinase inhibitors, which inhibit TRK receptors’ signaling, reverse multidrug resistance by directly inhibiting the function of ABC transporters and enhancing the efficacy of conventional chemotherapeutic drugs in different models of cancers [71].

NGF, through TRKA activation, can alter the expression of several molecules associated with cancer development and progression [58,72]. There is also evidence that NGF could control the expression of microRNAs (miRs) [72]. In EOC, deregulation in expression has been described, including alterations of the miR-200 family, cluster-17-92, and miR-23b, among others [73]. Some miRs could be associated with NGF/TRKA activation, modifying protein levels needed for EOC progression.

### 3.2. Role of microRNAs in EOC

The miRs belong to a group of non-coding RNA ranging between 18 to 25 nucleotides, involved in post-transcriptional regulation of messenger RNAs (mRNAs). They promote gene silencing through mRNA degradation or preventing the translation of proteins involved in critical cellular processes [74]. These mRNA post-transcriptionally promote gene silencing through mRNA degradation or prevent the translation of proteins involved in critical cellular processes [74]. The miRs are synthesized from long primary miR (pri-miR) in a process carried out by RNA polymerase II from precursor genes located in different parts of DNA [75]. The pri-miR are cleaved in the nucleus, resulting in precursor miR (pre-miR) being exported from the nucleus to the cytoplasm by exportin-5 [75]. In the cytoplasm, pre-miRs are enzymatically cleaved, releasing a miRNA duplex. This duplex comprises two strands. One of the strands corresponds to the “passing” strand, which can be degraded. The other strand is the mature miRNA, which will bind to the target mRNA, regulating the expression post-transcriptionally of proteins involved in processes such as proliferation, migration, and invasion [76]. 

In this way, deregulation of miRs can lead to the development of different pathologies, including cancer [77]. It has been identified that many miRs are deregulated in EOC compared to non-cancer ovarian cells [73]. Some miRs that increase their levels during the progression of this pathology are miR-200, miR-182, and miR-21 [78,79,80]. In contrast, miR-493, miR-23b, and miR-145 can be found within the miRs that decrease their levels [81,82,83]. Some dysregulated miRs in EOC are shown in Table 2, also shows the processes in which these miRs participate.

Upon binding to its high-affinity receptor, TRKA, NGF activates the MAPKs and PI3K/Akt signaling pathways, inducing cell survival, proliferation, migration, and invasion [77]. Interestingly, NGF stimulation decreases miR-23b and miR-145 in EOC cells, both oncosuppressor miRs [72]. These miRs decrease the translation of several oncogenic proteins, including the EMT protein ZEB-1 and the efflux transporter ABCB1 [85]. Therefore, the decrease of these miRs by NGF could increase these oncoproteins and increase the chemoresistance of EOC cells. 

Nonetheless, it was observed that when EOC cells are stimulated with NGF, they decrease miR-145 levels, suggesting that NGF regulates miR-145 in EOC cells [72]. Through its interaction with the TRKA receptor, it was proposed that NGF decreases transcription of miR-145 levels in EOC cells and increases oncogenic proteins involved in proliferation, migration, and angiogenesis [72]. Additionally, recent evidence suggests that decreased miR-145 levels would favor increased proteins related to chemoresistance, such as Multidrug Resistance protein 1 (MDR1) [85,86]. The miR-145 has been proposed as a suppressor of drug resistance in different cancers, regulating this resistance through different mechanisms and target mRNAs. In the case of EOC, studies have shown that miR-145 negatively regulates the ABCB1 transporter, also known as multidrug MDR1 or P-glycoprotein (P-gp). ABCB1 is related to resistance to paclitaxel by increasing the flow or expulsion of chemotherapeutics; so, it is suggested that miR-145 may regulate sensitivity to paclitaxel in EOC cells through drug accumulation [85].

In this respect, a decrease in miR-145 leads to an increase in the Programmed death-1 ligand (PD-L1) in a mechanism mediated by c-Myc, which induces immunological tolerance through the apoptosis of T cells helping tumor cells evade the immune system. Cisplatin provokes a rise in the PD-L1 level. The miR-145/c-Myc/PD-L1 axis contributes to resistance to cisplatin in EOC [86,87]. The above mentioned is consistent with an in silico analysis previously carried out by our group, which observed that miR-145 could decrease the expression of c-Myc, which was demonstrated with in vitro studies [72].

In addition, another chemoresistance mechanism in which miR-145 is participating is in the regulation of apoptosis. Several studies have indicated the relationship between miR-145 and the Bcl-2 family of proteins regulating apoptosis via the mitochondrial [86]. Our laboratory group determined by in silico analysis that Bcl-2 is a target of miR-145 [72]. In esophageal squamous cell carcinoma and cell lung cancer, overexpression of miR-145 led to a decrease in Bcl-2, the anti-apoptotic protein, and an increase in BAX in proapoptotic protein, caspase 3 excised and inducing apoptosis [88,89]. 

## 4. Mitochondria in Epithelial Ovarian Cancer: The Importance of Oxidative Phosphorylation, Chemoresistance, and miR-145 on NGF Regulation

### 4.1. Importance of Oxidative Phosphorylation in Epithelial Ovarian Cancer

In OC, the tumor develops in the same place, the abdominal cavity, which becomes a particular TME characterized by ascites and scarcity of oxygen and nutrients. Under these conditions, cancer cells shift to OXPHOS, which produces higher levels of ATP and help their survival in cancer cells. Therefore, OXPHOS could be an ideal metabolic target for EOC. The reports have shown that aerobic glycolysis adaptation does not involve a complete closure of OXHPHOS in tumors [90]. Recently, it was reported that in OC, chemoresistant cell types significantly depend on OXPHOS and, therefore, high sensitivity to OXPHOS inhibitors [91]. The authors further indicated that the preservation of functional mitochondria in EOC could be due to an augmented mitochondrial turnover rate, suggesting mitophagy inhibition as a potential strategy to tackle cisplatin-resistant OC progression [91].

Recent evidence shows mitochondrial deregulation or mitochondrial dysfunction in tumorigenesis processes that mirror alterations in biogenesis, morphology, and mitochondrial dynamics’ protein levels. In EOC, the increased mitochondrial number is associated with increased PGC1-α, Transcription factor A mitochondrial (TFAM) protein levels, and mtDNA content [41]. 

Signorile and collaborators reported that the OC mitochondria presented an augmented maximum length, and a decreased cristae width and junction diameter were associated with increased OPA1 and Prohibitin 2 (PBH2) protein levels. The modification of mitochondrial structure was associated with an increased level of OPA1 and PHB2 proteins. The authors reported an increased level of PHB2 related to an increased level of OPA1, thus representing another element of resistance to apoptosis in ovarian cancer cells [41]. 

Newly, Bindra and coworkers reported that gene expression of mitochondrial proteins in samples of ovarian tumors was associated with survival in women [92]. Additionally, the authors demonstrated that ovarian tumors showed higher mitochondrial content and ETC enzymatic activities than non-cancerous tissues. However, differences in mtDNA levels were not found. Interestingly, ETC activity was associated with interleukin-6 levels in samples with ascites. The authors proposed altered mitochondrial functional phenotypes in ovarian tumors [92]. 

### 4.2. Role of Mitochondria in Chemoresistance

The chemotherapy for EOC usually consists of treatment with carboplatin and paclitaxel to eradicate cancer cells but leaves behind cell populations that can cause the disease to reoccur [49]. Cancer patients with disseminated metastases have a lower response to chemotherapy treatment and, therefore, a lower probability of survival. Chemoresistance in OC is due to ovarian cancer cells’ intrinsic ability to resist chemotherapy and TME [49]. Effective treatment of EOC remains a significant challenge. 

The metabolic differences revealed that resistant cells to apoptosis undergo a shift toward OXPHOS [49]. Generally, the changes are accompanied by a reorganization in the mitochondrial network through increasing the mitochondrial ultrastructure [48]. The TME conditions with limited oxygen and glucose restrict metabolic plasticity in cancer cells, which has recently played a critical role in cancer progression and chemoresistance.

Zampieri and collaborators reported that chemosensitive EOC cell lines (A2780 and PEO1) displayed a glycolytic phenotype. Moreover, their chemoresistant counterparts (C200 and PEO4) exhibited a highly metabolically active phenotype with the ability to switch between OXPHOS or glycolysis [91]. The authors aimed to understand why cisplatin-resistant ovarian cancer cells had fitter mitochondria than cisplatin-sensitive ovarian cancer cells. Therefore, the authors proposed that resistant cells could recycle mitochondria faster in cisplatin-resistant ovarian cancer cells [91]. 

On the other hand, the direct impact of cisplatin on mitochondria induces ROS production that dictates the cell fate of ovarian cancer cells. The authors found that cisplatin-sensitive HGSOC cell lines contain higher mitochondrial content and levels of mitochondrial ROS (mtROS) than cells resistant to cisplatin-induced cell death [93]. The authors reported that mitochondria are in two ways critical for cisplatin sensitivity with knock-down of BAX/BAK, glutathione, and ROS scavenger, which reduced cisplatin-induced apoptosis [93].

Recently, Wang studied the regulation by Galectin-3 (Gal-3), a carbohydrate-binding protein related to cell migration, cell adhesion, and cell–cell interaction in cancer cells [94]. Gal-3 has been implied in the tumor progression and chemoresistance of EOC cells. Results demonstrated that overexpression of Gal-3 reduces apoptosis in the cisplatin-treated OVCAR-3 cells. In conclusion, Gal-3 diminishes the sensitivity of ovarian cancer cells to cisplatin via regulating cisplatin-induced mitochondrial dysfunction [94]. 

Dar and collaborators examined cellular bioenergetic profiling in EOC cell lines and demonstrated differences between sensitive and chemosensitive EOC cells [95]. Interestingly, A2780 and PEO1 (chemosensitive EOC cells) showed a glycolytic phenotype. In comparison, C200 and PEO4 (chemoresistant EOC cells) exhibited a highly metabolically active phenotype with the ability to switch between OXPHOS or glycolysis. The authors demonstrated that the chemosensitive cancer cells could not survive glucose deprivation, while the chemoresistant cells displayed adaptability. Thus, EOC cells seem to show the possibility of using glycolysis or OXPHOS as an energy source. The authors proposed that the flexibility for using OXPHOS or glycolysis may indicate an adaptation to achieve a higher “cellular fitness” that may also be associated with chemoresistance [95].

Lately, it has been reported that mitochondrial chaperone protein HSP60 is associated with decreased overall survival of EOC patients. The researchers determined if targeting HSP60 could induce cytotoxicity in sensitive and chemoresistant ovarian cancer cells, whether synergistic when combined with chemotherapeutic drugs [96]. Basal HSP60 mRNA levels were increased in chemoresistant EOC cells as compared with their sensitive counterparts. Moreover, the combination of the HSP60 antibody with cisplatin was significantly synergistic in both sensitive and chemoresistant EOC cells. The authors identified a novel target that may serve not only for OC treatment but also for sensitization of patients to chemotherapy. In this respect, the role of mitochondria in chemoresistance could be a new therapeutic approach for cancer therapy to decrease adverse effects and strengthen anti-tumor efficacy.

### 4.3. Interplay between NGF/TRKA and miR-145 Levels: Possible Implications for Mitochondria

Recent studies have established that miR-145 overexpression regulates mitochondrial metabolic reprogramming in ovarian cancer cells by decreasing ADP-ribosylation factor-like 5B (ARL5B), resulting in decreased mtDNA copy number, ATP levels, membrane potential, and mitochondrial markers, along with an increase in Cyt C, leading to the inhibition of mitochondrial function [97]. It has been reported that the Cyt C release is involved in apoptosis intrinsically in ovarian cancer cells [97,98].

Considering that NGF induces cellular proliferation and an anti-apoptotic condition in EOC cells [37], our research group demonstrated that NGF decreases the transcriptional activity and cellular levels of miR-145, which may increase mRNA and expression of Bcl-2, which leads to the inhibition of the intrinsic apoptotic pathway and the survival of the EOC cell (Figure 2). However, the implications that NGF would have on mitochondrial dynamics are not entirely clear. In HGSOC, the increase of DRP1, one protein involved in mitochondrial fission, as well as in cell survival and resistance to platinum compounds of OC cells, has been described [50]. Interestingly, Bcl-2 would be mediating fission mitochondrial and inducing apoptosis through DRP1 [99]. This finding is in agreement with recent studies of OC cells where the inhibition of DRP1 by mitochondrial division inhibitor 1 (Mdivi-1) inhibited mitochondrial fission and intrinsic apoptosis [100]. 

However, this apparent contradiction seems to have an explanation in the tumor environment. In hypoxic OC cells, an increase in mitochondrial fission and also resistance to cisplatin therapy occur. Nevertheless, when DRP1 and mitochondrial fission are inhibited in hypoxic OC cells, sensitization to therapy occurs, which allows determining that mitochondrial dynamics also depend on the tumor microenvironment [101].

## 5. Possible Therapeutic Strategies

Various strategies have been generated to target mitochondria for cancer therapies, including agents that aim for ETC, apoptotic pathways, ROS homeostasis, or mtDNA. New strategies to interfere with OXHPHOS should be considered for the treatment of ovarian tumors [102]. Many of the mitochondrial inhibitors are antibiotic or anti-parasitic drugs. Antibiotics such as macrolides, clindamycin, tetracycline, and linezolid cause an inhibitory effect on the expression of ETC complexes. For instance, salinomycin inhibits growth and decreases viability with paclitaxel [103]. Alone, salinomycin has an apoptotic effect that selects for ovarian cancer cells but not normal ovarian cells and is more potent against cisplatin-resistant ovarian cancer cell lines [104]. 

Our laboratory has investigated the use of metformin as an antitumoral agent. Metformin is a drug widely used to treat metabolic disorders [105], and has shown direct and indirect anti-tumor effects in several kinds of cancer cells, including EOC cells [106], both depending on or independently of the activation of the AMP-dependent kinase (AMPK) [106,107,108], their most known target at the cellular level. In patients with OC, metformin intake has been associated with decreased incidence and mortality [82,83,84,85,86,87], suggesting that metformin could be a promising drug for treating OC. Metformin restricts tumor growth by inhibiting specific steps in the mitochondrial ETC. The inhibition of mitochondrial respiratory chain complex I by metformin can reduce oxygen consumption and induce cytotoxicity [18,106]. Emerging evidence suggests that metformin could be an exciting candidate as a complementary therapy for EOC. 

Recent studies have shown that metformin could block the effect of NT in EOC cells. For instance, NGF/TRKA mediated proliferative and angiogenic effects in EOC cells. Molecularly, the treatment of EOC cells with metformin decreases the expression of c-MYC, β-catenin, and VEGF induced by NGF/TRKA [109]. In addition, metformin blocks the NGF-induced increase in MYC and β-catenin/TCF-Lef transcriptional activity in EOC cells, which involves an inhibition of the MAPK/ERK signaling pathway [106]. These results are consistent with in silico studies that showed that neoadjuvant metformin at clinically relevant dosages effectively treats OC [109]. Results of a phase II trial showed a better-than-expected overall survival in patients taking metformin [109]. On the other hand, metformin treatment could be beneficial to decrease metastasis potential. Chemoresistance tumors from patients treated with metformin had a 2.4-fold decrease in markers of cancer stem cells and increased sensitivity to cisplatin ex vivo [106] and a decreased immunodetection of the oncoproteins survivin, c-Myc, and β-catenin was found in patients’ biopsies.

One different mechanism proposed to explain metformin’s pleiotropic effects is the regulation of post-transcriptional mechanisms [110]. It is described that metformin treatment of EOC cells increases oncosuppressor miRs, such as miR-145 and miR-23b [109]. The downregulation of these miRs has been associated with an increase of proteins involved in chemoresistance as the drug-efflux transporter MDR1 and EMT proteins as ZEB1 in different models of cancer cells [111,112]. In EOC cells, metformin treatment reverts the NGF-mediated decrease of miR-145 and miR-23b, producing an upregulation of these miRs. The decrease of their target proteins’ miR regulation could partially explain metformin’s diverse and multiple antitumoral effects [109]. Hence, targeting tumor metabolism might represent a promising strategy to eradicate the cancer cells responsible for tumor relapse, especially when combined with classical chemotherapeutic drugs.

## 6. Conclusions

Nowadays, the mitochondria could act as a target to activate apoptosis, prevent cancer relapse, and avoid chemoresistance. Evidence shows that NGF stimulation produces an increase in the Bcl-2/BAX ratio, which supports the anti-apoptotic effects of NGF in EOC cells. 

It should be noted that mitochondria in chemoresistant EOC cells show a greater inclination towards OXPHOS than chemosensitive EOC cells that are primarily glycolytic, which can block the complexes that make up the ETC.

## 7. Future Perspectives

The mitochondria offer new alternatives that could act as a target for OC therapies. More studies are required to take advantage of the convenience related to activating the pathways involved in apoptosis. In addition, the mitochondria could be a sure target when avoiding chemoresistance and causing cancer cells to die. Probably, future studies could point to using conventional therapies plus some inhibitors of ETC and/or OXPHOS in ovarian cancer cells.

## Figures and Tables

**Figure 1 life-12-00008-f001:**
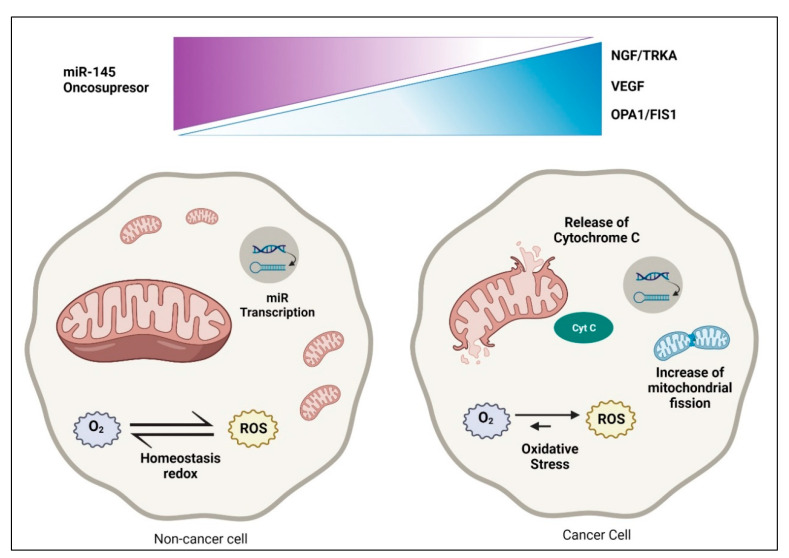
Significant metabolic changes in the epithelial ovarian cancer cell. Dysregulation in cancer cells compared with non-cancer cells. In the non-cancer cell, there is homeostasis redox, regulation between oncosuppressor and mitochondria, and various functions in dynamic equilibrium. There is augmented ROS and Cytochrome C (Cyt C) release in cancer cells, increased mitochondrial fission, and decreased transcription of miR-145.

**Figure 2 life-12-00008-f002:**
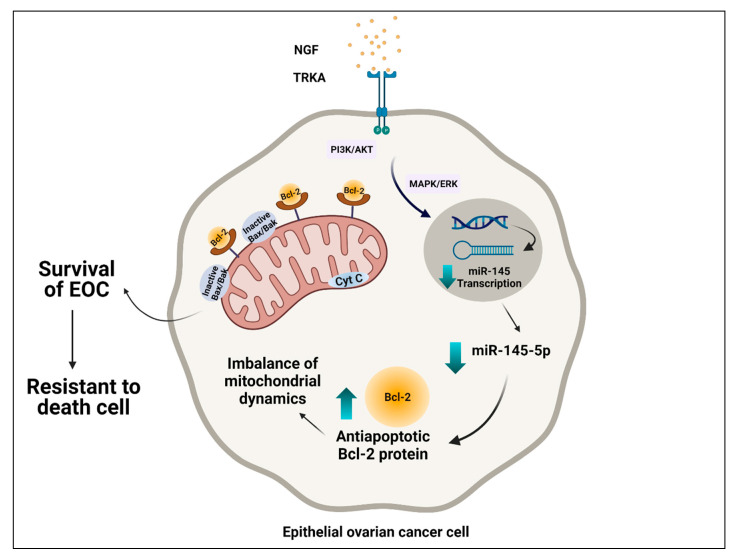
Possible interplay between mitochondria and regulation of NGF/TRKA and miR-145. Epithelial ovarian cancer cells are intrinsically resistant to cell death. Through its interaction with the TRKA receptor, NGF decreases transcription of miR-145 levels, causing an increase in oncogenic proteins involved in processes such as proliferation, migration, and angiogenesis. Additionally, NGF induces a decrease of transcriptional activity and cellular levels of miR-145, with an increase of the Bcl-2, expression of which leads to the inhibition of the intrinsic apoptotic pathway and the survival of the EOC cell.

**Table 2 life-12-00008-t002:** Expression of miRNAs and some processes they regulate in epithelial ovarian cancer.

miRs	Regulation	Sample	Process	Regulation Process	References
miR-200	↑	Blood (serum)	EMT, Metastasis	↑	[78]
miR-182	↑	Cell lines and tissue	Proliferation and invasion	↑	[79]
miR-21	↑	Cell lines	Proliferation and invasion	↑	[80]
miR-493	↓	Cell lines	Apoptosis (extrinsic and intrinsic)	↓	[81]
miR-23b	↓	Cell lines and tissue	Proliferation, Migration, and invasion	↑	[83,84]
miR-145	↓	Cell lines and tissue	Proliferation, Migration, and invasion	↑	[72,82]

Note: ↑ up-regulated expression and ↓ down-regulated expression.

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
