# Peer review of "Role of Mitochondria in Interplay between NGF/TRKA, miR-145 and Possible Therapeutic Strategies for Epithelial Ovarian Cancer"

_life, 2021, doi:10.3390/life12010008_

Round 1

Reviewer 1 Report

Daniela B. Vera et al give an exciting review about miRNA in Epithelial Ovarian Cancer (EOC). EOC is one of the most lethal tumors in women, because symptoms of EOC are hard to be seen and current therapy is not effective. They focused on the mitochondrial roles in EOC at first and gave information about microRNAs (such as miR-200, miR-182 etc.) in EOC. Their group has shown the roles of miR-145, and NGF on mitochondrial function. NGF-miR145-Bcl-2 axis leads to increased-cell proliferation, migration and chemotherapy-resistance on EOC.

This review is very well written and gets great insights of EOC development mechanism and new therapeutic strategy.

The authors should check this manuscript again, because there are some typos (For example, in line108, there is an unnecessary dot).

In addition, the authors summarized the miRNAs on EOC in Chapter 3. Table of miRNAs (function or targets of microRNAs, inducer or repressor of microRNAs, increased or decreased-expression at EOC etc.) could be helpful for readers to understand the roles of microRNAs in EOC.

Author Response

Reviewer #1

Daniela B. Vera et al. give an exciting review about miRNA in Epithelial Ovarian Cancer (EOC). EOC is one of the most lethal tumors in women because symptoms of EOC are hard to be seen, and current therapy is not effective. They focused on the mitochondrial roles in EOC at first and gave information about microRNAs (such as miR-200, miR-182, etc.) in EOC. Their group has shown the roles of miR-145, and NGF on mitochondrial function. NGF-miR145-Bcl-2 axis leads to increased-cell proliferation, migration and chemotherapy-resistance on EOC.

  • This review is very well written and gets great insights of EOC development mechanism and new therapeutic strategy.

Thank you very much for your comments; they certainly helped us to improve the manuscript. Regarding your suggestions, we receive them and make all the changes in the manuscript that we upload to the journal's platform. Next, I will point out where you can find the modifications made for us.

  • The authors should check this manuscript again, because there are some typos (For example, in line108, there is an unnecessary dot).

The manuscript was revised, and grammar mistakes were corrected.

  • In addition, the authors summarized the miRNAs on EOC in Chapter 3. Table of miRNAs (function or targets of microRNAs, inducer or repressor of microRNAs, increased or decreased-expression at EOC etc.) could be helpful for readers to understand the roles of microRNAs in EOC.

The table was incorporated into the manuscript on line 256. Please see the new Table 2 “Expression of miRNAs and some processes they regulate in epithelial ovarian cancer”. The table includes microRNA, increased or decreased expression, sample and/or cell line, the process that regulates, and references.

Reviewer 2 Report

This is a nice review demonstrating tumor metabolism as a promising target to combat epithelial ovarian cancer chemoresistance and improve patient survival.  

Major Comments:

  1. The authors should include the current chemotherapy regimens and its limitations in introduction section.
  2. The section “Metabolic changes in epithelial ovarian cancer cells” is very primeval. The authors should comprehensively discuss metabolic changes associated with epithelial ovarian cancer from in vitro, pre-clinical and clinical studies. High-grade serous ovarian cancers exhibit metabolic heterogeneity (PMID: 30244973). The authors should highlight the metabolic differences between chemo-sensitive vs. chemo-resistant, and more aggressive vs. less aggressive epithelial ovarian cancers. Adding a table highlighting metabolic changes would be more explanatory.
  3. Figure 2: The authors showed that increased mitochondrial fission leads to EOC cells survival and chemoresistance. Others have demonstrated positive correlation between mitochondrial fission and ovarian cancer cells apoptosis. Mitochondrial fission inhibitor-1 (Mdivi-1) however reduced apoptosis by decreasing Cyt C and caspase activity (PMID: 28963947). Similarly, high level of phospho-Drp1Ser637, which attenuate mitochondrial fission correlated with high-grade serous carcinoma (PMID: 32448194; also as cited by authors). Interestingly, significance of mitochondrial fission in cisplatin resistance ovarian cancer cells has also been shown (PMID: 31409904). The authors should include these studies to discuss the role of mitochondrial dynamics in EOC and chemoresistance.
  4. The role of tumor microenvironment and intercellular mitochondria exchange contributing tumor stem cell phenotype, improved metabolism and chemoresistance should be discussed.  
  5. The authors should check Grammar throughout the manuscript.

Author Response

Reviewer #2

This is a nice review demonstrating tumor metabolism as a promising target to combat epithelial ovarian cancer chemoresistance and improve patient survival.

Major Comments:

  1. The authors should include the current chemotherapy regimens and its limitations in introduction section.

Thanks to the reviewer for the comments and suggestions. About this point, some information on chemotherapy regimens and limitations were incorporated in lines 36-40 and 45-47.

  1. The section "Metabolic changes in epithelial ovarian cancer cells" is very primeval. The authors should comprehensively discuss metabolic changes associated with epithelial ovarian cancer from in vitro, pre-clinical and clinical studies. High-grade serous ovarian cancers exhibit metabolic heterogeneity (PMID: 30244973). The authors should highlight the metabolic differences between chemo-sensitive vs. chemo-resistant, and more aggressive vs. less aggressive epithelial ovarian cancers. Adding a table highlighting metabolic changes would be more explanatory.

The necessary antecedents were added in the section “Metabolic changes in epithelial ovarian cancer cells” (lines 98 to 140). Also, the section was complemented with table 1 about Metabolic changes in different models of epithelial ovarian cancer.

  1. Figure 2: The authors showed that increased mitochondrial fission leads to EOC cells survival and chemoresistance. Others have demonstrated positive correlation between mitochondrial fission and ovarian cancer cells apoptosis. Mitochondrial fission inhibitor-1 (Mdivi-1) however reduced apoptosis by decreasing Cyt C and caspase activity (PMID: 28963947). Similarly, high level of phospho-Drp1Ser637, which attenuate mitochondrial fission correlated with high-grade serous carcinoma (PMID: 32448194; also as cited by authors). Interestingly, significance of mitochondrial fission in cisplatin resistance ovarian cancer cells has also been shown (PMID: 31409904). The authors should include these studies to discuss the role of mitochondrial dynamics in EOC and chemoresistance.

The figure 2 was modified for shows imbalance of mitochondrial dynamics and the references were added.

  1. The role of tumor microenvironment and intercellular mitochondria exchange contributing tumor stem cell phenotype, improved metabolism and chemoresistance should be discussed.

The role of cancer stem cells (CSCs) in the tumoral microenvironment and their implications in ovarian cancer chemoresistance were added from lines 166 to 172.

The authors should check Grammar throughout the manuscript.

The grammar was checked and reviewed

Reviewer 3 Report

Life

COMMENTS TO THE EDITORS AND THE AUTHORS

Manuscript ID Life-1439183 “Role of mitochondria in interplay NGF/TRKA, miR-145

and possible therapeutic strategies for Epithelial Ovarian Cancer“

Dear Editors and the Authors,

Please find enclosed the comments for the above-mentioned manuscript.

A SUMMARY OF THE CONTENT

The authors stated that the review aims to discuss the role of mitochondria in the interplay between nerve growth factor (NGF), tropomyosin kinase A and miR-145 and possible therapeutic strategies that may decrease the mortality for epithelial ovarian cancer (EOC). The expression of NGF and TRKA, its high-affinity receptor increase during the progression of EOC by upregulation of oncogenic proteins vascular endothelial growth 16 factor (VEGF) and c-Myc. The expression of most oncoproteins is regulated by miRNAs. The effect of NGF/TRKA depends on the regulation of miRNA-145 levels in EOC and the mitochondria could be the target to activate cell death pathways, prevent cancer relapse, and prevent chemoresistance. Moreover, mitochondria in cisplatin-resistant EOC cells show a greater inclination towards OXPHOS compared to chemosensitive and primarily glycolytic EOC cells.

THE OVERALL OPINION OF THE MANUSCRIPT

The manuscript is within the scope of the journal and describes the important topic of life. The text is comprehensive but very easy to follow. The topic is of interest to general readers. The authors have publications in the field. They reference both, pioneered results. The text and the conclusions support the title.

A few minor suggestions are listed below.

(1) Please include recent relevant and high quality publications published in 2021 and related to the title of the manuscript.

(2) Please “rewrite” the abstract and/or conclusions since the significant number of sentences are identical.

(3) Please include one short paragraph dedicated to “Future perspectives” after “Conclusions”.

(4) Please use official abbreviations for all genes/proteins/signaling molecules etc.

Good luck and all the best :)

Author Response

Reviewer #3

Dear Editors and the Authors, Please find enclosed the comments for the above-mentioned manuscript.

A SUMMARY OF THE CONTENT: The authors stated that the review aims to discuss the role of mitochondria in the interplay between nerve growth factor (NGF), tropomyosin kinase A and miR-145 and possible therapeutic strategies that may decrease the mortality for epithelial ovarian cancer (EOC). The expression of NGF and TRKA, its high-affinity receptor increase during the progression of EOC by upregulation of oncogenic proteins vascular endothelial growth 16 factor (VEGF) and c-Myc. The expression of most oncoproteins is regulated by miRNAs. The effect of NGF/TRKA depends on the regulation of miRNA-145 levels in EOC and the mitochondria could be the target to activate cell death pathways, prevent cancer relapse, and prevent chemoresistance. Moreover, mitochondria in cisplatin-resistant EOC cells show a greater inclination towards OXPHOS compared to chemosensitive and primarily glycolytic EOC cells.

THE OVERALL OPINION OF THE MANUSCRIPT: the manuscript is within the scope of the journal and describes the important topic of life. The text is comprehensive but very easy to follow. The topic is of interest to general readers. The authors have publications in the field. They reference both, pioneered results. The text and the conclusions support the title. A few minor suggestions are listed below.

  • Please include recent relevant and high-quality publications published in 2021 and related to the title of the manuscript.

Thanks to the reviewer for the suggestion. New references from the year 2021 related to the subject and title were incorporated in the manuscript.

  • Please "rewrite" the abstract and/or conclusions since the significant number of sentences are identical.

The conclusions were reformulated in line 459.

  • Please include one short paragraph dedicated to "Future perspectives" after "Conclusions."

The section about future perspectives was incorporated from line 468.

  • Please use official abbreviations for all genes/proteins/signaling molecules etc.

All official names of genes and proteins were reviewed and corrected.

Round 2

Reviewer 2 Report

There are no comments to be noted.